# Application of the E-Nose as a Non-Destructive Technique in the Early Detection of *Monilinia laxa* on Plum (*Prunus domestica* L.)

**DOI:** 10.3390/s25247576

**Published:** 2025-12-13

**Authors:** Ana Martínez, Alejandro Hernández, Patricia Arroyo, Jesús S. Lozano, Alberto Martín, María de Guía Córdoba

**Affiliations:** 1Nutrición y Bromatología, Escuela de Ingenierías Agrarias, Universidad de Extremadura, 06007 Badajoz, Spain; anamd@unex.es (A.M.); amartin@unex.es (A.M.); mdeguia@unex.es (M.d.G.C.); 2Instituto Universitario de Investigación en Recursos Agrarios (INURA), Universidad de Extremadura, Avd. de la Investigación, 06006 Badajoz, Spain; jesuslozano@unex.es; 3Escuela Técnica Superior de Ingenieros Industriales, Universidad de Extremadura, 06006 Badajoz, Spain; parroyoz@unex.es

**Keywords:** E-nose, volatile profile, early decay, plum, *Monilinia laxa*

## Abstract

**Highlights:**

**Abstract:**

This study investigated the ability of an electronic nose system (E-nose) to detect early signs of fungal contamination in the red plum variety ‘Black Splendor’. We focused on identifying changes in volatile organic compounds (VOCs) that occur with decay. For this purpose, we compared two groups of plums: a control group (healthy plums) and a group inoculated with *Monilinia laxa*. VOCs from both groups were analyzed and quantified using gas chromatography/mass spectrometry (GC/MS). In parallel, E-nose signals were recorded at two key moments of fungal development: an early and an intermediate phase. The results revealed a strong correlation between E-nose signals and the aromatic profile characteristic of fungal contamination in plums. Linear discriminant analysis (LDA) models, developed from the E-nose data, achieved 100% differentiation between healthy and infected samples. Furthermore, these models discriminated with 100% accuracy between healthy plums and those with incipient contamination. These findings demonstrate that E-nose technology serves as a reliable, non-destructive approach for real-time assessment of plum quality throughout storage.

## 1. Introduction

Spain is the main stone fruit producer, with a production in the 2024 campaign of 1,789,644 tons, where plum production accounted for 9.1% of the total [1]. The fruit industry faces one of its greatest challenges: fungal contamination that causes significant losses both before harvest (pre-harvest) and after (postharvest). In the case of stone fruit, such as plums, the fungal genus *Monilinia* is one of the main causes of these losses.

This genus produces a disease called ‘brown rot’, with *Monilinia laxa*, *Monilinia fructicola* and *Monilinia fructigena* being the main culprits. Among the three species, *M. laxa* accounts for the highest incidence (approximately 85–90%), whereas *M. fructigena* represents a smaller proportion (around 10–15%) [2]. *M. fructicola*, however, was introduced in Europe, spreading to Spain, Italy, Switzerland or Slovakia at the beginning of the 21st century [3,4,5]. The species *Monilinia fructicola* was first detected in North America in 1883. Since then, its presence has been recorded in various countries, including California (1936), Canada (1976), Mexico (1999), as well as Guatemala and Panama (in 1976 and 1999, respectively). It has also been reported in several South American countries such as Argentina, Bolivia, Brazil, Peru, Venezuela (1976), and more recently in Ecuador, Paraguay, and Uruguay (1999) [6]. This was due to the faster spreading speed of *M. fructicola* because of its ability to produce more spores than the other two species [7]. Despite this, it has been found that this species is more prone to develop strains resistant to synthetic fungicides such as benzimidazole, dicarboximide and triazole [8].

*M. laxa* continues to be a primary contributor to losses in both stone and pome fruit crops. With the growing demand for high-quality fruit and stringent restrictions on fungicide application, brown rot caused by *M. laxa* poses a significant challenge to stone fruit production across Europe [9]. Consequently, the development of sustainable management strategies is essential to mitigate spoilage-related losses. Alternative non-destructive methods are increasingly being sought to allow rapid and early detection. These include techniques such as NIR spectroscopy, hyperspectral imaging (HSI) and electronic noses, which offer a solution for on-line detection of fruit quality and spoilage, allowing mitigation of fungal impact on fruit production. These techniques have also been found to enable quantification of each quality parameter, as well as sorting of fruits according to colour, shape, size and texture [10].

The electronic nose (E-nose) represents a novel technology that provides a rapid and cost-efficient solution with minimal response time. It is composed of four main components: a gas sensor array, a sampling unit, a data acquisition module, and a data processing system [11,12]. Recent advancements have significantly enhanced its capabilities, enabling broad applications within the food industry. Thus, its effectiveness has been demonstrated for product classification, identification, monitoring, control and traceability [13]. Tyagi et al. [14] classified fruits such as apple, banana, orange, grape and pomegranate into three ripening stages (unripe, ripe and overripe). They established, with the help of neural networks, an aromatic fingerprint of these different stages, achieving a discrimination with an accuracy of more than 95%. This technology also allows the detection of sensory defects in foods such as shoe, butyric and musty odor defects in table olives as a consequence of abnormal fermentations [15].

One of the most important applications in the food industry is the ability for early detection of fungal contamination, which is of vital importance during pre-harvest, postharvest and/or storage [16]. This capability has been studied extensively. For example, Jia et al. [17] were able to distinguish between fresh and moldy apples inoculated with *Penicillium expansum* and *Aspergillus niger*. Similarly, Gu et al. [18] were able to discriminate between healthy and *Aspergillus*-contaminated rice batches on the second day, achieving a success rate of 92.86%. In addition, gas chromatographic analysis showed that the volatile compounds were closely related to the fungal species present in the rice.

There are many other studies such as the one presented by Pan et al. [19], which analyzed the volatile compounds emitted by control strawberries and strawberries inoculated with *Botrytis* spp., *Penicillium* spp. and *Rhizopus* spp. for 10 days by gas chromatography–mass spectrometry (GC-MS) and E-nose. The findings indicated that by the second day, differentiation between control and infected strawberries was achieved with a classification accuracy of 96%. Likewise, Haghbin et al. [20] successfully detected early contamination by *Botrytis* spp. in kiwifruit, reaching a discrimination rate exceeding 90%. Collectively, these studies underscore the adaptability of E-nose systems for identifying diverse fungal pathogens across various food matrices, reinforcing their value within the fruit and vegetable sector. Multivariate analysis techniques such as Principal Component Analysis (PCA), Linear Discriminant Analysis (LDA), Partial Least Squares Regression (PLSR) and machine learning algorithms, including Support Vector Machines (SVMs) and Artificial Neural Networks (ANNs), were used in most of the studies found. These techniques are essential for extracting meaningful information from the complex datasets generated by chemical sensor arrays [21,22,23].

Despite the promising results of electronic noses, several challenges remain in their application to fungal detection in fruit. These include the complexity and variability of volatile organic compounds (VOCs) influenced by fruit ripeness and environmental conditions, the limited selectivity of metal oxide sensors, and the need for robust multivariate models to interpret high-dimensional data. Although previous studies have focused mainly on pathogens such as *Botrytis* or *Penicillium* in other fruit matrices, research on *Monilinia laxa* in plums remains scarce, and most of the available literature focuses on other stone fruits such as peaches. This gap highlights the need for specific approaches to improve early detection in this fruit.

Our work aims to approach these challenges by identifying specific VOC biomarkers associated with *Monilinia laxa* infection. Special focus was placed on the relationship between these VOCs and the metal oxide (MOX) sensors of an electronic nose (E-nose). The final objective was to determine whether this portable device is a non-destructive and viable technique for the early detection of *Monilinia laxa* on plum.

## 2. Materials and Methods

### 2.1. Experimental Design

For this study, we selected *Monilinia laxa* CA1 [24] from the Culture Collection of the CAMIALI group research (UEx (University of Extremadura)). This strain was grown on potato dextrose agar (PDA) at 25 °C for 10 days.

We used plums of the ‘Black Splendor’ variety (*Prunus domestica* L.), known for their red flesh. These were purchased from a supermarket between July and August 2023, although they are usually harvested at the end of June. To ensure freshness, the plums were kept refrigerated at 1 °C for a maximum of two days. Prior to the experiment, they were sanitized with 70% ethanol and artificially inoculated to ensure reproducibility and controlled infection onset. Subsequently, 3 mm wide by 3 mm deep wounds were made using a sterilized tip. Inoculation was carried out by placing a small cube of mycelium (3 × 3 mm) in each wound. This allowed the establishment of two experimental batches (inoculation factor): (i) batch with inoculated fruits (M_ batch), and (ii) control batch, with uninoculated fruits.

Once inoculated, plums were packed in transparent polyethylene containers (25 cm wide × 25 cm high × 11 cm deep) and kept at a temperature of 25 °C. For the VOC study, three samples were taken from each lot at two key times, defined by the progress of infection (stage factor): (i) early stage with 18 mm diameter damage (at 2 days post-inoculation) and (ii) intermediate stage with 37 mm damage (at 4 days post-inoculation). In addition, a batch of uninoculated plums was also included. This experimental design was replicated three times to ensure the reliability of the results.

### 2.2. Volatile Compound Analysis

#### 2.2.1. Volatile Extraction

To capture volatile organic compounds (VOCs), samples were placed in 780 cm^3^ glass bottles with plastic lids. To eliminate any residual VOCs from prior samples, the headspace was flushed with high-pressure GC-grade air for two minutes [22,25].

Volatile compounds were collected by inserting an HS-SPME fiber (Supelco, Supelco, Bellefonte, Pennsylvania, EE. UU.). directly through the bottle cap. The extraction of VOCs was carried out at 25 °C for 40 min, followed by gas chromatography–mass spectrometry (GC/MS) analysis as outlined in Section 2.2.2. To ensure the reliability of the results, each analysis was performed in triplicate, and the whole experiment was repeated three times. It is important to note that, prior to their first use, the fibers were activated according to the conditioning instructions.

#### 2.2.2. Gas Chromatography/Mass Spectrometry (GC/MS) Analysis

For VOC identification and quantification, the procedure described by Serradilla et al. [26] was employed using an Agilent 6890 GC system coupled to a 5973 mass spectrometer (Agilent Technologies, Little Falls, DE, USA). Separation of compounds was achieved on a 50 m × 0.32 mm ID capillary column with a 1.05 μm film thickness, coated with 5% phenyl and 95% polydimethylsiloxane (HP-5MS equivalent, model 19091J-215, Agilent Technologies, Madrid, Spain). Compound identification relied on Kovats retention indices calculated with n-alkanes (R-8769, Sigma Chemical Co., St. Louis, MO, USA), and mass spectra were compared against the NIST/EPA/NIH library, requiring a minimum match quality of 90%. To confirm specific compounds, retention times and spectra were cross-checked with an in-house library generated from pure standards under identical conditions. Quantification was performed by integrating peak areas from total ion current chromatograms to estimate relative abundance.

### 2.3. E-Nose Analysis

#### 2.3.1. E-Nose System

The electronic nose (E-nose) employed in this study was developed by the Sensory Systems research group at the University of Extremadura [27]. It incorporates 11 commercial metal oxide (MOX) sensors, providing a wide detection range (Appendix A). These sensors include the following: (i) Bosch BME680, which measures environmental parameters such as temperature (°C), pressure (hPa), relative humidity (%RH), and gas resistance (Ω); (ii) Sensirion SGP30, capable of detecting equivalent CO_2_ concentration (eCO_2_), total volatile organic compounds (TVOCs), and raw resistance values for hydrogen and ethanol; (iii) ScioSense CCS811 and (iv) iAQ-Core, both of which measure eCO_2_, TVOC, and sensor resistance.

Each sensor is integrated into a compact module featuring analogue conditioning circuits, analog-to-digital converters, a microcontroller, communication interfaces, and a heated microboard. The system operates on a 3.7 V lithium battery and transmits data wirelessly via Bluetooth to a dedicated mobile application.

#### 2.3.2. Measurement Process and Data Analysis

E-nose analyses were performed in parallel with VOC determinations to maintain consistency across experimental conditions. Each electronic nose reading was performed in two sequential phases of 120 s each: an Adsorption Phase, where the sensors were directly exposed to the headspace of the samples for adsorption of odorant molecules, and a Desorption Phase, where the sensors were exposed to clean air to establish a baseline. Each measure was derived from five consecutive readings interspersed with blank intervals. In total, 72 measurements were performed, distributed among the different batches established.

To derive a representative value for each measurement cycle, a baseline adjustment algorithm was applied. This characteristic value was computed as the proportional difference between a reference baseline (*Vref*) and the reading obtained during volatile exposure (*Vodor*). The baseline (*Vref*) corresponded to the mean of the last five measurements recorded while the sensor was exposed to ambient air, whereas the volatile value (*Vodor*) was defined as the mean of the last five readings taken during exposure to the sample headspace. The final value (*Vf*) was determined using the following equation:*Vf* = (*Vref* − *Vodor*) × 100^−1^

### 2.4. Statistical Analysis

Statistical analyses were performed using SPSS software version 21.0 for Windows (SPSS Inc., Chicago, IL, USA). Initially, descriptive statistics were applied to characterize the volatile organic compounds (VOCs) detected. A two-way analysis of variance (ANOVA) was then conducted to evaluate the influence of inoculation and developmental stage on VOC profiles. Principal component analysis (PCA) was employed to examine the discriminatory potential of volatile compounds among samples.

Furthermore, Pearson correlation coefficients were calculated, and hierarchical cluster analysis (HCA) was implemented to assess the effectiveness of E-nose sensors in reflecting plum volatile profiles. In HCA, sample distances were computed using squared Euclidean metrics, and clustering was performed via the agglomerative intergroup linkage method. To classify plum batches by storage stage, linear discriminant analysis (LDA) was applied using a stepwise algorithm based on Wilks’ lambda to select the most relevant variables for stage differentiation, both across the full dataset and in early stages. Cross-validation was incorporated into LDA models to rigorously evaluate predictive performance and minimize overfitting, thereby enhancing robustness and generalizability. Finally, the most effective LDA model was visualized by projecting sample groups within the space defined by the discriminant functions (DF).

## 3. Results and Discussion

### 3.1. Volatile Compounds

A total of 62 volatile organic compounds (VOCs) emitted by red-fleshed plums were identified in this study. These compounds were classified into the following chemical families: alcohols (7), esters (47), aldehydes (4), hydrocarbons (2), ketones (1), and carboxylic acids (1) (Table 1). Esters emerged as the predominant group, with the major contributors being butyl hexanoate (v51) (29.87%), ethyl hexanoate (v29) (20.06%), butyl butanoate (v28) (17.64%), and butyl acetate (v12) (5.65%). Esters are primarily responsible for the floral and fruity aroma notes in fruits and have been consistently reported as key aromatic compounds in various plum cultivars [28,29,30]. This aromatic profile is not exclusive to plums but is also characteristic of other *Prunus* species, such as peaches and nectarines [31,32,33].

Among the 47 esters identified, the majority (25 VOCs) remained relatively stable regardless of the presence of *M. laxa*. However, distinct trends were observed in the remaining esters. Compounds such as methyl acetate (v1), n-propyl acetate (v4), ethyl 2-methyl butanoate (v13), pentyl acetate (v23), and butyl hexanoate (v51) exhibited increased concentrations in inoculated samples, suggesting a possible pathogen-induced biosynthesis or release. Conversely, compounds like 2-methylbutyl propanoate (v27), isobutyl pentanoate (v36), 2-methylbutyl butanoate (v38), and pentyl pentanoate (v40) showed reduced levels in infected batches, with most also declining during storage (Table 1).

Fungal contamination in stone fruits induces notable alterations in the volatile organic compound (VOC) profile, which can significantly impact both aroma and overall fruit quality. These changes are typically driven by pathogen metabolism, host defense responses, and degradation processes triggered by infection [34,35,36]. Previous studies have reported elevated ester concentrations following fungal infection, often resulting in more pronounced fruity, floral, and toasty aroma notes. For instance, Fanesi et al. [37] linked compounds such as propyl acetate, ethyl acetate, pentyl acetate, ethyl 2-methylbutanoate, and butyl acetate to peaches infected with *Monilinia* spp. Conversely, other infections may reduce the levels of desirable aroma compounds, leading to a flatter and less appealing sensory profile [38].

Regarding storage stages (Ps), although most esters did not show significant variation, some compounds exhibited notable changes. Specifically, butyl propanoate (v22), butyl 2-methylpropanoate (v25), 2-methylpropyl butanoate (v26), ethyl hexanoate (v29), 2-methylbutyl butanoate (v38), and methylbutanoate (v54) decreased during storage. In contrast, methyl acetate (v1), methyl butanoate (v5), butyl butanoate (v28), ethyl heptanoate (v41), and 4-pentenyl hexanoate (v58) showed increased levels (Table 1 and Appendix A).

The second most abundant chemical family was alcohols, accounting for approximately 1.3% of the total AAU. As shown in Table 1, most alcohols increased in plums inoculated with *M. laxa*, with 2-methylbutan-1-ol (v7) being the only compound significantly affected by the storage stage, showing a marked increase over time. Fungal infections such as *Monilinia* spp. (brown rot) or *Botrytis cinerea* (grey mold) are known to alter the volatile profile of stone fruits, often leading to elevated levels of fermentation-related volatiles due to tissue degradation and microbial activity [36,39].

The remaining compounds—aldehydes, hydrocarbons, ketones, and carboxylic acids—collectively accounted for less than 1% of the total AAU. Although these compounds did not exhibit significant quantitative changes, their potential contribution to the overall aroma profile should not be overlooked. Some of them may possess low olfactory thresholds, meaning even small concentrations could influence sensory perception. However, due to their limited abundance and lack of consistent variation, they are considered less relevant for electronic nose (E-nose) applications focused on detecting fungal contamination or assessing fruit quality.

Following the two-way ANOVA, a Principal Component Analysis (PCA) was conducted to further explore the multivariate structure of the dataset. PCA enables the visualization of overall patterns and relationships among VOCs, facilitating the identification of compound clusters and sample groupings based on their volatile profiles. Figure 1B shows a principal component analysis where the first component explains 65.7% of the variance. We can observe that the controls in the different stages (C_C, C_1S and C_2S) are on the negative axis of PC_1 and the *Monilinia* batches on the positive axis, thus obtaining a clear separation. In addition, the second principal component, which explains 15.1% of the variance, separates nectarines inoculated with *M. laxa* on the second day (M_1S) from nectarines inoculated with *Monilinia* on the fourth day (M_2S), being located on the positive and negative axis of PC_2, respectively.

This differentiation is also reflected in the distribution of the volatile compounds produced during the experiment. Thus, in Figure 1A, we can see that compounds such as methyl acetate (v1), methyl butanoate (v5), 2-methylbutan-1-ol (v7), ethyl butanoate (v11), ethyl 2-methylbutanoate (v13), ethyl pentanoate (v21) or ethyl heptanoate (v41) are associated with the middle infection stage and compounds such as n-propyl acetate (v4), 3-hexanol (v14), 1-hexanol (v17), propyl butanoate (v20) or (Z)-3-hexenyl acetate (v31) are associated with the early infection stage (M_1S). In addition, we can observe that the controls in both infection stages are separated from day 0, associated with volatiles such as octanal (v30) or 2-methylbutyl hexanoate (v57).

This allows us to conclude that infection by *M. laxa* significantly alters the profile of volatile compounds emitted by stored plums. This alteration makes it possible to clearly distinguish between healthy and infected fruit, and even to differentiate according to storage time. These results underline the great potential of the electronic nose (E-nose) for early detection of the presence of *Monilinia* in plums during their storage period.

### 3.2. Relationship Between VOCs and Signals from E-Nose Sensors

As mentioned above, the electronic nose has a great applicability in the food industry, however, the large volume of data it generates requires techniques that can group this information based on the similarity of sensor responses. One of them is Hierarchical Cluster Analysis (HCA) [40], which, together with other sources of information such as volatile compound analysis, allows us to improve the accuracy of such clustering or classification [41,42].

In this context, our study presents in Table 2 a comprehensive overview of the correlation patterns between 26 volatile organic compounds (VOCs) and the response signals of 11 metal oxide (MOX) sensors. The VOCs are grouped into four clusters based on their correlation profiles, revealing distinct sensor sensitivity patterns and potential chemical affinities.

The largest group, Cluster 1, comprises 19 VOCs, including 14 esters and 5 aldehydes. This cluster exhibited high sensitivity and strong positive correlations with sensors M_1_ (RBME680), M_4_ (H_2_SGP30), M_5_ (EthanolSGP30), and M_8_ (ResohmCCS811), while showing negative correlations with sensors M_2_ (CO_2_SGP30), M_3_ (TVOCSGP30), M_6_ (CO_2_CCS811), M_7_ (TVOCCCS811), M_9_ (CO_2_iAQ), and M_10_ (TVOCiAQ). Notable compounds such as methyl acetate, methyl butanoate, isobutyl acetate, 3-methylbutyl acetate, methyl hexanoate, ethyl heptanoate, butyl (Z)-3-hexenoate, and 1-hexanol demonstrated high sensitivity across most sensors (*p* < 0.05), with the direction of correlation—positive or negative—depending on the specific sensor (Table 2). A previous study by Martínez et al. [22] described similar sensor responses to VOCs emitted by molds, where esters showed positive correlations with RBME680, H_2_SGP30, EthanolSGP30, ResohmCCS811, and RiAQCore.

Cluster 3 comprises five VOCs, including butyl propanoate, octanal, and pentyl pentanoate, which exhibit moderate and variable correlations. These compounds show positive responses primarily from sensors M_2_ (CO_2_SGP30), M_3_ (TVOCSGP30), M_6_ (CO_2_CCS811), M_7_ (TVOCCCS811), M_9_ (CO_2_iAQ), and M_10_ (TVOCiAQ)—precisely the sensors that displayed negative correlations with the compounds in Cluster 1. This inverse response pattern suggests that the VOCs in Cluster 3 elicit a sensor profile opposite to that of Cluster 1, highlighting the differential sensitivity of the MOX array to distinct compound structures.

Cluster 2 comprises a single compound, hexanal, which did not exhibit any significant correlation with the MOX sensors. Similarly, Cluster 4 contains only one compound, (Z)-3-hexenyl acetate, which showed a weak positive correlation exclusively with sensor M_11_ (RiAQCore). This absence or minimal sensor activity can be attributed to its low concentration or reduced chemical reactivity with the MOX array.

Overall, the clustering reveals two dominant sensor response profiles: one characterized by strong positive correlations with M_1_ (RBME680), M_4_ (H_2_SGP30), M_5_ (EthanolSGP30), and M_8_ (ResohmCCS811) (Cluster 1), and another with moderate positive responses from M_2_ (CO_2_SGP30), M_3_ (TVOCSGP30), M_6_ (CO_2_CCS811), M_7_ (TVOCCCS811), M_9_ (CO_2_iAQ), and M_10_ (TVOCiAQ) (Cluster 3). The clear separation between these profiles highlights the differential selectivity of the MOX sensors and supports their potential use in VOC classification and spoilage detection.

### 3.3. Determination of Incipient Fungal Decay of Peaches by E-Nose During Postharvest Storage

To evaluate the postharvest behavior of plums during storage and to compare the different experimental batches, Linear Discriminant Analysis (LDA) models were developed using data obtained from an electronic nose. The predictive performance of these models was assessed through the leave-more-out cross-validation technique. Classification was based on the results of Principal Component Analysis (PCA) of the volatile compounds, which enabled a clear distinction between two groups—healthy plums (control) and plums contaminated with *M. laxa*—as well as among three groups: healthy plums, plums infected with *M. laxa* at the initial stage of fungal growth (M_1S), and plums infected with *M. laxa* at a more advanced stage of fungal development (M_2S). The results of the Linear Discriminant Analysis (LDA) are presented in Table 3 and Table 4, which list the most relevant sensors ranked from highest to lowest importance according to the stepwise selection algorithm applied. These tables also report the number and percentage of correct classifications obtained during both the calibration and prediction phases, providing a clear indication of the model’s effectiveness.

As presented in Table 3, the sensors CO_2_CCS811, TVOCCCS811, ResohmCCS811, and TVOCSGP30 achieved complete discrimination between healthy plums and those exhibiting early-stage contamination, with a 100% correct classification rate. Likewise, CO_2_CCS811, TVOCCCS811, and H_2_SGP30 successfully differentiated healthy fruits from those infected with *M. laxa*, also reaching 100% accuracy. These distinctions among the respective batches are further supported by the results illustrated in Figure 2.

Furthermore, the sensors CO_2_SGP30, CO_2_CCS811, TVOCCCS811, CO_2_iAQ, and H_2_SGP30 demonstrated the ability to differentiate healthy plums from those infected with *M. laxa* at distinct stages of fungal development (M_1S and M_2S), with a 100% correct classification rate (Table 4; Figure 3).

These results highlight the high effectiveness of LDA models based on electronic nose sensor data in accurately discriminating between different sample groups. For instance, Martínez et al. [22] achieved over 97% accuracy in distinguishing healthy Golden Delicious apples from those with incipient *Penicillium expansum* contamination. Similarly, Rezaee et al. [43] reported 90% accuracy in classifying pistachios contaminated with *Aspergillus flavus* at various storage stages.

## 4. Conclusions

This study provides pioneering evidence of the applicability of E-nose technology for non-destructive, real-time monitoring of red-fleshed plum quality during storage, specifically in detecting early fungal contamination by *M. laxa*. While previous research has demonstrated the efficacy of E-nose systems in identifying spoilage in other commodities such as rice and kiwifruit [18,20], the present work is among the first to successfully discriminate between healthy and incipiently contaminated “Black Splendor” plums with 100% accuracy using Linear Discriminant Analysis (LDA). This underscores the novelty of the approach and its potential to fill a critical gap in postharvest quality control of stone fruits. The strong correlation between sensor signals and aromatic changes associated with spoilage highlights the robustness of the method. Given its capacity for early detection, the E-nose presents a promising tool for integration into agro-industrial workflows, enabling timely interventions and reducing postharvest losses. Future research should prioritize reducing detection time, conducting external validation on naturally infected commercial fruit under refrigerated storage conditions, and implementation of advanced machine learning models, such as artificial neural networks (ANNs), to enhance predictive performance and facilitate broader adoption across the supply chain.

## Figures and Tables

**Figure 1 sensors-25-07576-f001:**
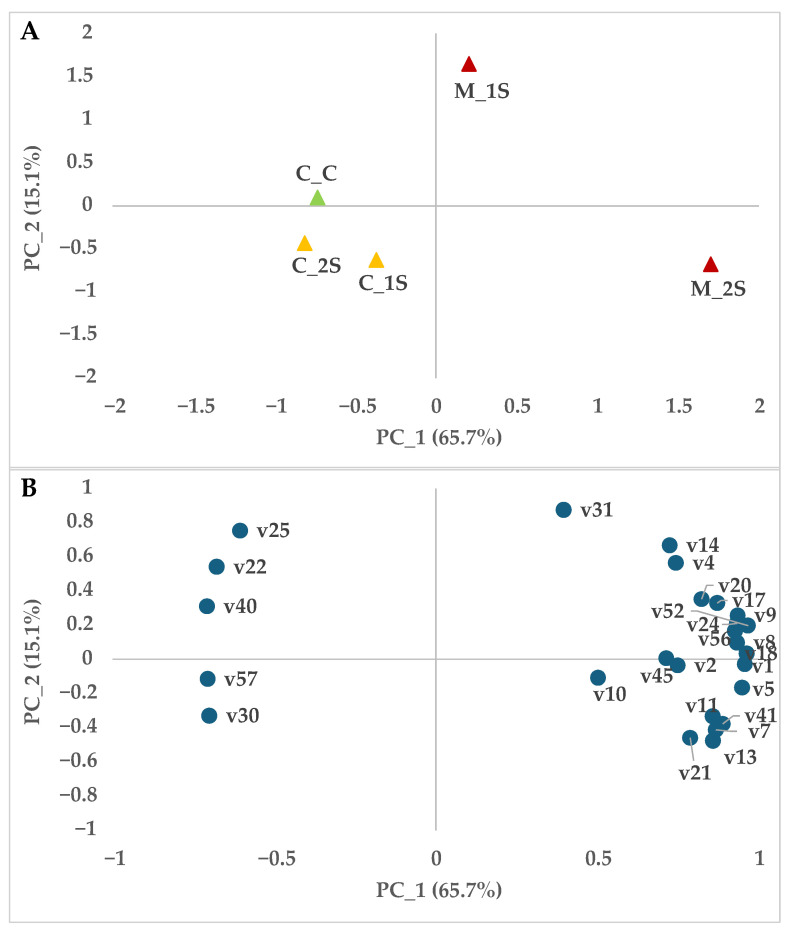
Loading plot (**A**) and Score plot (**B**) after principal component analysis of volatile compounds of the plum samples in the planes defined by the two first principal components (PC1 and PC2). Control samples Day 0 (C_C; ▲); Uninoculated samples (▲); Inoculated samples (▲); Early stage (M_S1) and Middle stage (M_S2); Volatile compounds (code described in Table 1; ●).

**Figure 2 sensors-25-07576-f002:**
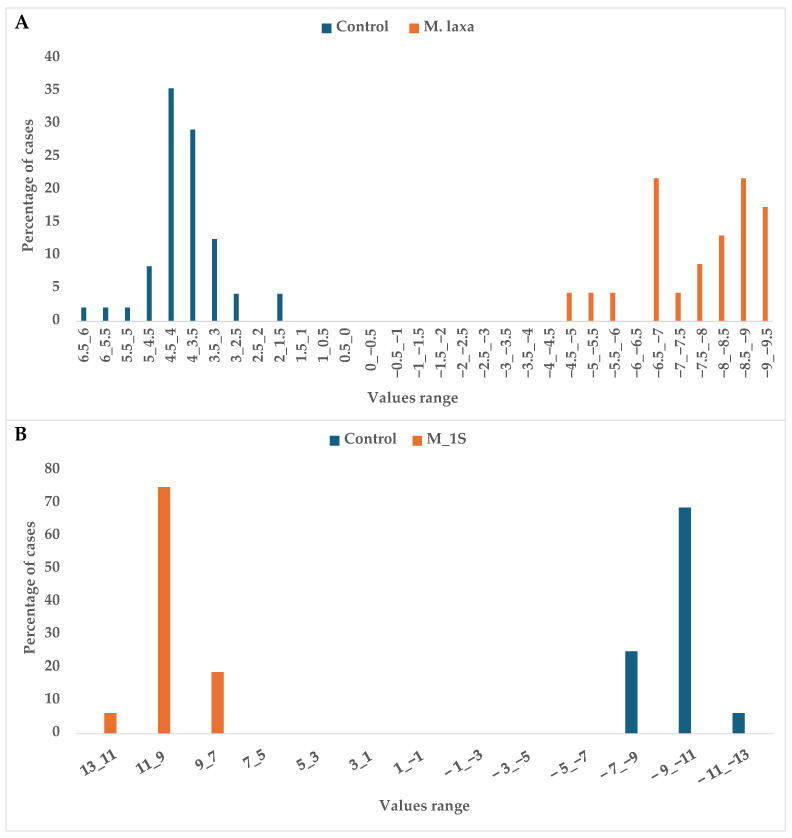
Histogram of plum samples grouped into control and inoculated batches on DF for all samples (**A**) and early stage (**B**).

**Figure 3 sensors-25-07576-f003:**
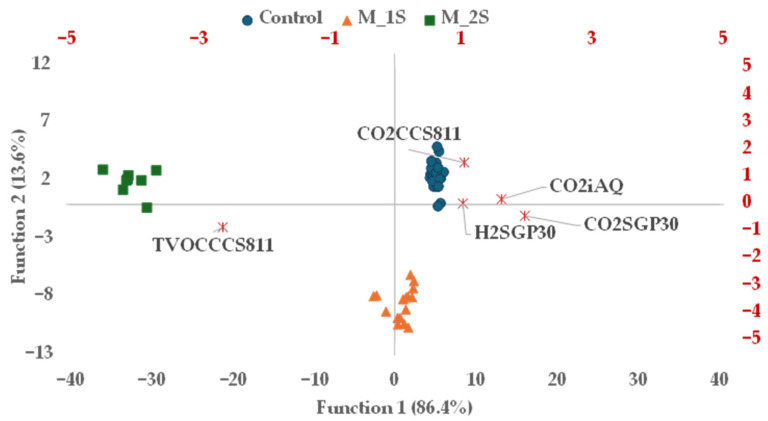
Plum samples grouped into batches: Control (uninoculated sound samples; ●), M_S1 (inoculated sample in early stage; ▲) and M_S2 (inoculated samples in middle stage; ■). Variable loadings (✳) projected on DF1 (Discriminant Function 1) and DF2 (Discriminant Function 2) plane (red scale).

**Table 1 sensors-25-07576-t001:** Volatile compounds identified in plum samples.

					Mean ^5^			*p* Values ^7^	
RT ^1^	CD ^2^	Volatile Compounds	ID ^3^	IK ^4^	AAU	%	RSD ^6^	Ps	Pi
	** *Alcohol* **				**9695**	**1**			
3.7	v6	3-methylbut-3-en-1-ol	B	727	214	0.03	87		+++
3.8	v7	2-methylbutan-1-ol	B	731	3535	0.49	134	+++	++
4.6	v8	1-pentanol	B	762	458	0.06	177		+++
8.2	v14	3-hexenol	C	890	579	0.08	142		
8.9	v17	1-hexanol	B	906	4742	0.66	100		+++
16.1	v33	Ethyl-1-hexanol	B	1033	107	0.01	138		
19.4	v45	Phenylethyl alcohol	B	1113	59	0.01	304		
	** *Ester* **				**698,007**	**98**			
1.7	v1	Methyl acetate	A	522	186	0.03	168	+	+++
2.2	v2	Ethyl acetate	A	616	5333	0.75	60		++
3.4	v4	n-propyl acetate	B	715	215	0.03	166		+++
3.5	v5	Methyl butanoate	B	719	515	0.07	165	+++	
4.9	v9	Isobutyl acetate	B	773	519	0.07	113		
5.8	v11	Ethyl butanoate	B	807	12,357	1.73	119		+
6.4	v12	Butyl acetate	B	828	40,421	5.65	75	++	++
8	v13	Ethyl 2-methylbutanoate	B	883	423	0.06	168	+	+++
9.3	v18	3-methylbutyl acetate	B	913	328	0.05	163		
9.4	v19	2-methylbutyl acetate	B	915	51	0.01	271		+++
10.3	v20	Propyl butanoate	B	929	1052	0.15	77		
10.4	v21	Ethyl pentanoate	B	931	243	0.03	156		
10.8	v22	Butyl propanoate	B	937	12,132	1.70	58	−−−	−
11	v23	Pentyl acetate	B	940	869	0.12	118		+++
11.5	v24	Methyl hexanoate	B	948	1734	0.24	136		
12.8	v25	Butyl 2-methylpropanoate	B	969	870	0.12	52	−−−	
12.9	v26	2-methylpropyl butanoate	B	971	3698	0.52	78	−−−	
13.6	v27	2-methylbutyl propanoate	B	982	104	0.01	246		−−−
14.8	v28	Butyl butanoate	B	1002	126,173	17.64	76	+++	+++
14.9	v29	Ethyl hexanoate	B	1005	143,501	20.06	92	−−−	−−
15.2	v31	(Z)-3-hexenyl acetate	C	1012	2754	0.38	120		++
15.5	v32	Hexyl acetate	B	1019	22,226	3.11	102	++	
16.6	v34	Butyl 2-methylbutyrate	B	1045	18,733	2.62	55	−−−	
16.8	v35	3-methybutyl butanoate	B	1050	143	0.02	157	−−−	
17.1	v36	Isobutyl pentanoate	B	1057	4	0.00	217	−−−	−−−
17.2	v37	3-methylbutyl isobutyrate	C	1060	851	0.12	78	−−−	+
17.3	v38	2-methylbutyl butanoate	B	1062	9329	1.30	78	−−−	−−−
17.6	v39	4-pentenyl butanoate	C	1069	206	0.03	174	++	
18.8	v40	Pentyl pentanoate	C	1097	18,381	2.57	36	−	−−−
19	v41	Ethyl heptanoate	C	1103	160	0.02	211	+++	
19.1	v42	3-methylbut-2-enyl butanoate	C	1105	2679	0.37	44		
19.3	v44	Hexyl propanoate	B	1110	1224	0.17	87		
20.5	v46	Pentyl 2-methylbutyrate	C	1141	284	0.04	164	−	
20.9	v47	Hexyl 2-methylpropanoate	C	1151	81	0.01	259		
21	v48	Isobutyl hexanoate	C	1154	4838	0.68	36		
21.2	v49	2-methylbutyl pentanoate	C	1159	109	0.02	196		
22.3	v50	(E)-3-hexenyl butanoate	C	1187	2464	0.34	82	−−	
22.6	v51	Butyl hexanoate	C	1195	213,711	29.87	60		+++
22.7	v52	Butyl (Z)-3-hexenoate	C	1197	3600	0.50	76		
24	v54	(3Z)-3-Hexenyl 2-methylbutanoate	C	1330	127	0.02	194	−−−	
24.1	v55	Hexyl 2-methylbutanoate	C	1335	3526	0.49	54		−−
24.6	v56	Isopentyl hexanoate	C	1357	294	0.04	148		+
24.7	v57	2-methylbutyl hexanoate	C	1361	10,714	1.50	61		−−−
25	v58	4-pentenyl hexanoate	C	1374	263	0.04	201	+++	
25.9	v59	Pentyl hexanoate	C	1403	5114	0.71	100		−
26.2	v60	3-methyl-2-butenylhexanoate	C	1406	566	0.08	192		
29.2	v62	Hexyl hexanoate	B	1438	24,900	3.48	164		
	** *Aldehyde* **				**957**	**0.1**			
5.7	v10	Hexanal	B	803	98	0.01	159		+
15	v30	Octanal	B	1007	206	0.03	117		−−
19.2	v43	Nonanal	B	1108	222	0.03	247		
23	v53	Decanal	B	1240	430	0.06	71		
	** *Hydrocarbons* **				**1120**	**0.16**			
8.4	v15	Ethylbenzene	C	898	130	0.02	209		
8.7	v16	p-xylene		903	990	0.14	147	−−	−−−
	** *Ketone* **				**3178**	**0.44**			
3.1	v3	3-pentanone	B	704	3178	0.44	74	−	−
	** *Carboxilic acid* **				**2492**	**0.35**			
29	v61	(E)-3-octenoic acid	C	1436	2492	0.35	96		

^1^ RT: Retention time (min). ^2^ CD: Code of volatile compound used in Figure 1 and Figure 2. ^3^ ID: Reliability of identification: A, identified by a comparison to standard compounds; B, tentatively identified by the NIST/EPA/NIH mass spectrum library (comparison quality > 90%) and Kovats index; C, tentatively identified by the NIST/EPA/NIH mass spectrum library (comparison quality < 90%). ^4^ KI: Kovats retention index. ^5^ AAU: Arbitrary Area Unit; (%): Relative percentage. ^6^ RSD: Relative Standard Deviation. ^7^ Ps: *p* values of stage factor; Pi: *p* values of inoculation factor. The significance of the effects is indicated by + (positive effect) or − (negative effect). One, two, and three symbols correspond to *p* values less than 0.1, 0.05, and 0.01, respectively.

**Table 2 sensors-25-07576-t002:** Cluster groups of volatile compounds according to Pearson correlation values between their area and the responses values of the different MOX used.

		MOX ^1,2^										
Cluster	VOCs	M_1	M_2	M_3	M_4	M_5	M_6	M_7	M_8	M_9	M_10	M_11
1	Methyl acetate	++	−−	−−	++	+	−−	−−	++	−−	−−	
1	Ethyl acetate			−	+							
1	n-propyl acetate	+		−	+	+	−		+			
1	Methyl butanoate	++	−−	−−	++	++	−−	−−	++	−−	−−	
1	2-methyl-1-butanol	++	−−	−−	++	+	−	−−	++	−	−	
1	1-pentanol	+	−−	−−	++	+	−−	−−	++	−	−	
1	Isobutyl acetate	++	−−	−−	++	++	−−	−−	++	−−	−−	+
1	Ethyl butanoate	+	−−	−−	+		−	−−	+	−	−	
1	Ethyl 2-methylbutanoate	+	−−	−−	+		−	−−	+	−	−	
1	3-hexenol	+			+	+	−−		++	−	−	++
1	1-hexanol	++	−	−−	++	++	−−	−−	++	−−	−−	+
1	3-methylbutyl acetate	++	−−	−−	++	+	−−	−−	++	−−	−−	
1	Propyl butanoate	++	−	−	+	+	−−	−	++	−	−	+
1	Ethyl pentanoate		−−	−				−				
1	Methyl hexanoate	++	−−	−−	++	++	−−	−−	++	−−	−−	++
1	Ethyl heptanoate	++	−−	−−	++	+	−−	−−	++	−−	−−	
1	Phenylethyl alcohol		−	−								
1	Butyl (Z)-3-hexenoate	++	−−	−−	++	++	−−	−−	++	−−	−−	+
1	Isopentyl hexanoate	++	−−	−−	++	+	−−	−−	++	−	−	
2	Hexanal											
3	Butyl propanoate		+	+				+				
3	Butyl 2-methylpropanoate		++	+				+				
3	Octanal	−			−	−	+		−	+	+	−
3	Pentyl pentanoate		+				+	+	−	+	+	
3	2-methylbutyl hexanoate						+	+	+	+	+	
4	(Z)-3-hexenyl acetate											+

^1^ MOX: RBME680 (1); CO2SGP30 (2); TVOCSGP30 (3); H2SGP30 (4); EtanolSGP30 (5); CO2CCS811 (6); TVOCCCS811 (7); ResohmCCS811 (8); CO2iAQ (9); TVOCiAQ (10); RiAQCore (11). ^2^ Positive (+) or negative (−) correlation (*p* < 0.01, −−− or +++; *p* < 0.05, −− or ++; *p* < 0.1, − or +).

**Table 3 sensors-25-07576-t003:** Performance in calibration and prediction of Linear Discriminant Analysis (LDA) classification applied to the plum samples grouped in two batches.

			Plum Batches		Total
Samples	Selected Variable	Count	Control	*M. laxa*	
Early store ^4^	CO2CCS811	n	16	16	32
	TVOCCCS811	Computed classes			
	ResohmCCS811	Class ^1^	16	16	32
	TVOCSGP30	% ^2^	100	100	100
		Predicted classes ^3^			
		Class ^1^	16	16	32
		% ^2^	100	100	100
All ^4^	CO2CCS811	n	48	24	72
	TVOCCCS811	Computed classes			
	H2SGP30	Class ^1^	48	24	72
		% ^2^	100	100	100
		Predicted classes			
		Class ^1^	48	24	72
		% ^2^	100	100	100

^1^ Number of cases correctly classified. ^2^ Percentage of cases correctly classified. ^3^ The classification outcomes obtained after applying the cross-validation procedure. ^4^ Early store: Samples included in the early infection stage (18 mm and 2 days post-inoculation); All: Samples included in the two established stages of infection: Early and Intermediate (37 mm and 4 days post-inoculation).

**Table 4 sensors-25-07576-t004:** Performance in calibration and prediction of Linear Discriminant Analysis (LDA) classification applied to the plum samples grouped in control and spoiled samples (early stage and middle stage).

		Plum Batches			Total
Selected Variable	Count	Control	M_1S	M_2S	
CO_2_SGP30	n	48	16	8	72
CO_2_CCS811	Computed classes				
TVOCCCS811	Class ^1^	48	16	8	72
CO_2_iAQ	% ^2^	100	100	100	100
H_2_SGP30	Predicted classes ^3^				
	Class ^1^	48	16	8	72
	% ^2^	100	100	100	100

^1^ Number of cases correctly classified. ^2^ Percentage of cases correctly classified. ^3^ The classification outcomes obtained after applying the cross-validation procedure.

## Data Availability

The raw data supporting the conclusions of this article will be made available by the authors on request.

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
