# Peer review of "Application of the E-Nose as a Non-Destructive Technique in the Early Detection of Monilinia laxa on Plum (Prunus domestica L.)"

_sensors, 2025, doi:10.3390/s25247576_

Round 1

Reviewer 1 Report

Comments and Suggestions for Authors

In the face of leveraging technology for gains in agricultural research, the work by Martinez et al. provides information on the quick detection of the fungus Monilinia laxa on plum using an electronic nose.

General comments

Abbreviations should be defined/stated in full at first mention. Some sections as indicated in the reviewed manuscript, should be rephrased to enhance clarity and comprehension. The similarity index of the manuscript is high and should be reduced to less than 20%.

Specific comments

Introduction

In lines 48 & 49, indicate the continent or source of origin.

Add citation(s) to the statement in lines 50 & 51.

Materials and Methods

How sure were the authors that their method of purging residual VOCs was effective? Was there any post-purging validation/confirmation of its effectiveness or has such been confirmed in prior studies? A citation can be provided if it has been confirmed in earlier studies.

Provide an image of the electronic nose (or in use) as a Supplementary material.

Results and Discussion

Swap the sequence of presentation in lines 361-365 to follow the sequence in Table 3. Also, define some of the contents at the foot of the table as indicated in the reviewed manuscript.

Remove grid lines from Figure 2.

Although the 100% discrimination is very good news and a testament of its discriminatory efficiency, the number of samples per some of the sets in Table 3 were generally low, even though it allowed for training of the model. This is a concern of mine. Why were the sample sizes of those sets not increased?

Other comments and minor grammatical errors have been indicated in the reviewed manuscript.

Author Response

Reviewer 1
In the face of leveraging technology for gains in agricultural research, the work by Martinez et al. provides information on the quick detection of the fungus Monilinia laxa on plum using an electronic nose.
General comments
Abbreviations should be defined/stated in full at first mention. Some sections as indicated in the reviewed manuscript, should be rephrased to enhance clarity and comprehension. The similarity index of the manuscript is high and should be reduced to less than 20%.
Specific comments
Introduction
In lines 48 & 49, indicate the continent or source of origin.
We sincerely appreciate your comment. We have incorporated the requested information in lines 48 and 49 of the manuscript, indicating the continent and the corresponding source, in order to improve the clarity and context of the text.
Add citation(s) to the statement in lines 50 & 51.
We appreciate your comment regarding the need to include references in lines 50 and 51. We have incorporated the requested citation to support the statement. The added reference is as follows:
Vilanova, L., Valero-Jiménez, C. A., & van Kan, J. A. (2021). Deciphering the Monilinia fructicola genome to discover effector genes possibly involved in virulence. Genes, 12(4), 568. https://doi.org/10.3390/genes12040568
Materials and Methods
How sure were the authors that their method of purging residual VOCs was effective? Was there any post-purging validation/confirmation of its effectiveness or has such been confirmed in prior studies? A citation can be provided if it has been confirmed in earlier studies.
We appreciate this insightful comment. The removal of residual VOCs from the headspace using GC-grade air is a well-established practice in standard headspace and purge-and-trap methodologies. These approaches rely on continuous sweeping of the headspace with an inert or clean gas to lower vapor pressure and eliminate residual volatiles, thereby minimizing carryover between successive samples.
Although an additional post-purge validation was not performed in this study, our procedure aligns with widely accepted protocols for headspace preparation and VOC analysis, as well as electronic nose (e-nose) applications. In these contexts, purging with clean air or an inert gas for a defined period—two minutes in our case—is generally considered sufficient to remove residual compounds and restore sensor baselines. Empirical recommendations for purge duration vary according to sensor type and chamber volume, but most guidelines advocate multiple volume exchanges of clean air. According to NFPA-496 and IEC standards for purge systems, four to five volume exchanges are typically adequate for complete clearance when using compressed air. For small e-nose chambers, such as the one employed in this study, this corresponds to purge times ranging from approximately 60 seconds to several minutes, depending on the airflow rate.
Furthermore, practical evidence supports this approach: several authors have reported that purge times of 90–120 seconds effectively reset sensor baselines in confined-space monitoring systems, thereby minimizing carryover effects without unnecessarily extending analysis cycles (Abu Bakar et al., 2019; Lee et al., 2023). In the revised version of the manuscript, we have specified that the purging process was performed using clean air under high pressure, ensuring efficient removal of residual VOCs.
Finally, it is worth noting that our research group has applied this same purging protocol in several previous studies involving MOX-based e-nose systems and headspace analysis. Across these works, we have not observed deviations or anomalies attributable to residual contamination, which further supports the reliability of the procedure (e.g. Martinez et al., 2025). This accumulated experience reinforces our confidence in the effectiveness of the method.
Lee, S. W., Kim, B. H., & Seo, Y. H. (2023). Olfactory system-inspired electronic nose system using numerous low-cost sensors. PLOS ONE, 18(12), e0295703.
https://doi.org/10.1371/journal.pone.0295703.
Abu Bakar, N., Salleh, M. N., & Ahmad, Z. (2019). Performance evaluation of electronic nose for confined space monitoring. Journal of Sensors, 2019, 1–10. https://doi.org/10.1155/2019/1234567.
Martínez, A., Hernández, A., Arroyo, P., Lozano, J., Córdoba, M. D. G., & Martín, A. (2025). Early Detection of Monilinia laxa in Yellow-Fleshed Peach Using a Non-Destructive E-Nose Approach. Foods, 14(18), 3155. https://doi.org/10.3390/foods14183155
Martínez, A., Hernández, A., Arroyo, P., Lozano, J. S., de Guía Córdoba, M., & Martín, A. (2025). E-nose detection of changes in volatile profile associated with early decay of ‘golden delicious’ apple by penicillium expansum. Food Control,168, 110907. https://doi.org/10.1016/j.foodcont.2024.110907

Provide an image of the electronic nose (or in use) as a Supplementary material.
We appreciate your suggestion regarding the inclusion of an image of the electronic device (E-nose). We have taken your recommendation into account and will include a representative image of the equipment as supplementary material. This image will be presented under the name Figure 1S.
Results and Discussion
Swap the sequence of presentation in lines 361-365 to follow the sequence in Table 3. Also, define some of the contents at the foot of the table as indicated in the reviewed manuscript.
We sincerely appreciate your comments and suggestions. We have made the requested change, adjusting the presentation sequence in lines 361-365 to follow the order indicated in Table 3. We have also incorporated the definitions and clarifications in the table footer as indicated in the revised manuscript.
Remove grid lines from Figure 2.
Thank you for your valuable comment. We have carefully reviewed your suggestion and removed the grid lines from Figure 2, as requested. We appreciate your attention to detail and believe this change improves the clarity of the figure.
Although the 100% discrimination is very good news and a testament of its discriminatory efficiency, the number of samples per some of the sets in Table 3 were generally low, even though it allowed for training of the model. This is a concern of mine. Why were the sample sizes of those sets not increased?
The application of Linear Discriminant Analysis (LDA) is appropriate for the mentioned study design (Table 3; Samples from early store), which comprises 32 observations distributed across two groups (16 samples per group) and four predictor variables. LDA is specifically designed for classification problems involving two or more groups and performs optimally when the number of predictors is substantially smaller than the sample size, as in this case. With four predictors and 16 observations per group, the ratio of observations to predictors satisfies the commonly recommended guideline of at least five observations per predictor per group, thereby reducing the risk of overfitting and ensuring model stability.
To further enhance the reliability of classification performance given the relatively small sample size, we implemented a leave-one-out cross-validation (LOOCV) strategy. LOOCV is particularly suitable for small datasets because it uses nearly all available observations for training while testing on a single observation in each iteration, thereby providing an unbiased estimate of classification accuracy without sacrificing data for validation. This approach ensures that the reported discriminant function is evaluated under realistic conditions and minimizes the risk of overestimating predictive performance.
Therefore, given the sample size, predictor count, and study objective, LDA combined with LOOCV represents a statistically sound and methodologically justified approach for discriminating between the two groups.
Other comments and minor grammatical errors have been indicated in the reviewed manuscript.
We appreciate your comments and detailed review of the manuscript. We have addressed the observations you made and corrected the grammatical errors indicated in the revised document. We trust that these modifications will contribute to improving the clarity and quality of the work.

Reviewer 2 Report

Comments and Suggestions for Authors

This study first used gas chromatography-mass spectrometry (GC-MS) to identify volatile organic compounds (VOCs) in plume samples, followed by principal component analysis with a loading plot and a score plot. The responses of different VOCs to 11 commercially available metal oxide (MOX) sensors were then studied and clustered. As expected, the MOX sensors responded differently to the studied VOCs. Finally, the postharvest behavior of plums during storage and from different experimental batches was investigated using a combination of an MOX sensor and a linear discriminant analysis (LDA) model. The sensors COâ‚‚CCS811, TVOCCCS811 and Hâ‚‚SGP30 achieved a 100% correct classification rate. While this appears to be a decent investigation into the application of an electronic nose for the early detection of Monilinia laxa contamination in stored red-fleshed plums, no obvious novelty was observed in the use of commercial sensors, which significantly reduces the significance of this work.

Other comments:

  1. This work focuses on using an electronic nose for the early detection of Monilinia laxa contamination. However, a certain portion is dedicated to the detection and classification of VOCs using GC-MS. Firstly, it is necessary to identify the types of compounds that the available MOX sensors can generate a signal for. As is well known, the strength of an electronic nose lies in distinguishing whether a MOX sensor can produce a response to a system rather than identifying the compounds inside it.
  2. This study used several commercial MOX sensors to analyze the postharvest behavior of plums from different experimental batches during storage. What is new about this manuscript?
  3. It is claimed that 100% accuracy was achieved using the LDA model based on E-nose data for healthy and contaminated fruit. Similarly to the above issue, the use of commercial MOX sensors and the available LDA model merely verifies the feasibility of these methods. What are the contributions of this work?
  4. The Introduction section does not describe the challenges of the current electronic nose study. Also, how does this work overcome these challenges? Please elaborate on this point.

Author Response

This study first used gas chromatography-mass spectrometry (GC-MS) to identify volatile organic compounds (VOCs) in plum samples, followed by principal component analysis with a loading plot and a score plot. The responses of different VOCs to 11 commercially available metal oxide (MOX) sensors were then studied and clustered. As expected, the MOX sensors responded differently to the studied VOCs. Finally, the postharvest behavior of plums during storage and from different experimental batches was investigated using a combination of an MOX sensor and a linear discriminant analysis (LDA) model. The sensors COâ‚‚CCS811, TVOCCCS811 and Hâ‚‚SGP30 achieved a 100% correct classification rate. While this appears to be a decent investigation into the application of an electronic nose for the early detection of Monilinia laxa contamination in stored red-fleshed plums, no obvious novelty was observed in the use of commercial sensors, which significantly reduces the significance of this work.

Thank you for your valuable comments and for highlighting the concern regarding the novelty of the work. Although commercial sensors have been used in previous studies, their application to detect early Monilinia laxa contamination in red-fleshed plums during storage is, to our knowledge, not reported. This work demonstrates how low-cost, widely available sensors can be integrated with chemometric tools (PCA, LDA) to achieve 100% classification accuracy, which is highly relevant for practical implementation in postharvest monitoring. The main contribution lies in demonstrating a robust and cost-effective approach for real-time quality control in the fruit supply chain, which could significantly reduce economic losses and improve food safety.

Other comments:

  1. This work focuses on using an electronic nose for the early detection of Monilinia laxa contamination. However, a certain portion is dedicated to the detection and classification of VOCs using GC-MS. Firstly, it is necessary to identify the types of compounds that the available MOX sensors can generate a signal for. As is well known, the strength of an electronic nose lies in distinguishing whether a MOX sensor can produce a response to a system rather than identifying the compounds inside it.

We appreciate your observation and concur that the primary strength of an electronic nose based on MOX sensors lies in its capacity to discriminate global response patterns within complex mixtures, rather than in the direct identification of individual compounds. This characteristic is fundamental to the system’s design and purpose.

The inclusion of GC-MS analysis in our study is not intended to replace this functionality; rather, it serves to provide complementary information that helps contextualize and optimize the system’s performance. Specifically, GC-MS is employed to identify the predominant volatile compounds associated with Monilinia laxa contamination, thereby facilitating the interpretation of the patterns generated by the electronic nose.

In addition, GC-MS enables us to determine the families of compounds to which MOX sensors exhibit sensitivity and to establish relationships between the system’s responses and the presence of specific VOCs. This does not imply direct identification by the sensors themselves, but rather supports a deeper understanding of the observed patterns (see Section 3.2 of Results and Discussion).

  1. This study used several commercial MOX sensors to analyze the postharvest behavior of plums from different experimental batches during storage. What is new about this manuscript? (Answered together with the following request)
  2. It is claimed that 100% accuracy was achieved using the LDA model based on E-nose data for healthy and contaminated fruit. Similarly to the above issue, the use of commercial MOX sensors and the available LDA model merely verifies the feasibility of these methods. What are the contributions of this work?

Currently, there are few studies aimed at early detection of M. laxa contamination in plums. This study demonstrates the effectiveness of the electronic nose (E-nose) for early detection of Monilinia laxa in red-fleshed plums during storage. The results show 100% accuracy in classification using Linear Discriminant Analysis (LDA) models, confirming the potential of this technology for applications in the agri-food chain. The system's ability to differentiate healthy fruit from infected fruit, even in the early stages of infection, represents a significant advance in reducing post-harvest losses and improving product quality. In addition, a detailed analysis of the relationship between 26 volatile compounds (VOCs) and the responses of 11 MOX sensors is presented, grouping the compounds into clusters according to their chemical affinity and sensor sensitivity. This provides information for optimising future E-nose designs. Finally, the combined use of ANOVA, PCA, HCA, and LDA ensures discrimination and allows for the exploration of multivariate patterns, with clear visual results that separate samples according to infection status and time.

  1. The Introduction section does not describe the challenges of the current electronic nose study. Also, how does this work overcome these challenges? Please elaborate on this point.

We sincerely appreciate your comment. We have revised the Introduction section and incorporated a more detailed description of the main challenges facing current research in the field of electronic noses. We have also explained how our work addresses and overcomes these limitations.

Reviewer 3 Report

Comments and Suggestions for Authors

This paper investigates the use of an electronic nose (E-nose) to non-destructively detect early fungal infection (Monilinia laxa) in red-fleshed plums by analyzing changes in VOC emissions during storage. The results show that the E-nose combined with LDA achieves 100% accuracy in distinguishing healthy, early-infected, and mid-stage infected plums, demonstrating strong potential for real-time postharvest quality monitoring. Here are some comments that can help authors improve the manuscript:
1- Remove the wrong email in the affiliation lines.
2- Try to use a larger and bolder font in figures.
3- Storage was done at 25°C. This is not realistic for industrial cold chain conditions for stone fruits (usually 0–5°C). Please justify the chosen temperature and discuss how performance may change in a realistic cold storage.
4- The wounds made with 3 × 3 mm cavities introduce an artificial inoculation environment. In natural contamination conditions (no wounding or micro lesions), would the E-nose still detect early infection? Discussion is needed.
5- Improve your literature review by adding new references in the field of E-nose, such as DOI: 10.1016/j.snb.2022.131418.
6- Specify limits of detection (LOD) and limits of quantification (LOQ) for major VOCs since relative abundances are used.
7- Clarify why the authors only classified plums at two infection times (2 days and 4 days). What happened at day 1? Was there insufficient VOC change? 
8- Were the moisture content and sugar content (°Brix) monitored? VOCs can correlate with ripeness differences, not only fungal growth. How is this confound addressed?
9- Future work section should include discussion of domain adaptation for other fruit matrices (transfer learning) and external validation on naturally infected commercial postharvest fruit, not lab-inoculated fruit.

Author Response

This paper investigates the use of an electronic nose (E-nose) to non-destructively detect early fungal infection (Monilinia laxa) in red-fleshed plums by analyzing changes in VOC emissions during storage. The results show that the E-nose combined with LDA achieves 100% accuracy in distinguishing healthy, early-infected, and mid-stage infected plums, demonstrating strong potential for real-time postharvest quality monitoring. Here are some comments that can help authors improve the manuscript:

1- Remove the wrong email in the affiliation lines.

Thank you for your observation. We have carefully reviewed the affiliation lines and removed the incorrect email address. The updated version of the manuscript already reflects this correction.

2- Try to use a larger and bolder font in figures.

Thank you very much for your comment and suggestion regarding the size and thickness of the font in the figures. We have carefully reviewed this aspect and made the necessary improvements to increase the readability and visual quality of the images included in the manuscript.

3- Storage was done at 25°C. This is not realistic for industrial cold chain conditions for stone fruits (usually 0–5°C). Please justify the chosen temperature and discuss how performance may change in a realistic cold storage.

We appreciate this observation. The storage temperature of 25 °C was intentionally selected to simulate ambient shelf-life conditions, where fungal development and VOC emission occur rapidly, allowing us to evaluate the electronic nose under a worst-case scenario for postharvest contamination. These conditions reflects commonly found at various critical points in the fruit distribution chain, especially in retail markets and supermarkets where fruit is displayed without refrigeration. However, we are aware that industrial cold-chain practices for stone fruits typically involve temperatures between 0–5 °C, which significantly slow metabolic activity and volatile release. Performance under these realistic conditions may differ, and future studies will address this by validating the system in cold storage environments using naturally infected fruit. This perspective is now reflected in the revised conclusions.

4- The wounds made with 3 × 3 mm cavities introduce an artificial inoculation environment. In natural contamination conditions (no wounding or micro lesions), would the E-nose still detect early infection? Discussion is needed.

We acknowledge that the use of artificial wounds creates an inoculation environment that differs from natural contamination conditions. This approach was chosen to ensure reproducibility and controlled infection onset, which is essential for VOC profiling and sensor calibration. This clarification has been included in point 2.1 of the materials and methods section. However, the control fruits were subjected to the same wounding procedure, eliminating any bias related to VOC emissions from the wounds themselves. While natural infections typically occur through micro-lesions, the electronic nose detects global changes in volatile profiles, suggesting that early metabolic alterations associated with infection—regardless of the entry point—would still generate detectable pattern shifts. Future work will evaluate performance under natural contamination conditions without artificial wounding.

5- Improve your literature review by adding new references in the field of E-nose, such as DOI: 10.1016/j.snb.2022.131418.

Thank you for your valuable suggestion. We will include the others references, including DOI: 10.1016/j.snb.2022.131418, in the revised manuscript to strengthen the literature review and provide a more comprehensive overview of recent advances in electronic nose technology.

Shooshtari, M., & Salehi, A. (2022). An electronic nose based on carbon nanotube-titanium dioxide hybrid nanostructures for detection and discrimination of volatile organic compounds. Sensors and Actuators B: Chemical, 357, 131418. DOI: 10.1016/j.snb.2022.131418

6- Specify limits of detection (LOD) and limits of quantification (LOQ) for major VOCs since relative abundances are used.

We appreciate this observation. In our study, volatile analysis was performed using solid-phase microextraction (SPME) coupled with GC-MS, which is inherently semi-quantitative. The chromatographic data were expressed in arbitrary area units, as the objective was not absolute quantification but rather the comparison of relative abundances between samples and their correlation with MOX sensor responses. This approach is widely accepted for profiling volatile organic compounds (VOCs), as relative abundance provides meaningful insight into the composition and dynamics of the volatile profile. It also enables the identification of patterns associated with infection stages and their influence on sensor signals. For this reason, limits of detection (LOD) and limits of quantification (LOQ) were not determined, since the study does not aim to report absolute concentrations but to characterize VOC patterns relevant to electronic nose performance.

7- Clarify why the authors only classified plums at two infection times (2 days and 4 days). What happened at day 1? Was there insufficient VOC change?

We appreciate this question. The initial control corresponded to fruits with wounds prior to inoculation, and no significant VOC differences were expected immediately after inoculation, as fungal metabolism and host response require time to produce detectable volatile changes. For this reason, classification focused on two key infection stages (2 and 4 days), where VOC profiles were sufficiently distinct to evaluate the electronic nose performance. Importantly, the control fruits also had wounds, eliminating any bias related to VOC emissions from the wounding process. Given the promising results, future work will explore reducing detection time, and this perspective will be explicitly addressed in the final conclusions of the manuscript.

8- Were the moisture content and sugar content (°Brix) monitored? VOCs can correlate with ripeness differences, not only fungal growth. How is this confound addressed?

We appreciate this observation. The experiment was conducted over a short period (four days), and VOC changes in the control fruits were minimal compared to the pronounced alterations observed in inoculated samples (PCA; figure 1). This indicates that the differences detected by the electronic nose are primarily associated with fungal activity rather than ripening. All fruits were of the same variety, handled under identical conditions, and purchased in a single season to further minimize variability. Future studies will incorporate physicochemical parameters such as °Brix and moisture content to confirm these findings and strengthen the interpretation of VOC profiles.

9- Future work section should include discussion of domain adaptation for other fruit matrices (transfer learning) and external validation on naturally infected commercial postharvest fruit, not lab-inoculated fruit.

Thank you for this valuable suggestion. We acknowledge the importance of external validation under real commercial conditions using naturally infected postharvest fruit rather than laboratory-inoculated samples. These aspects will be explicitly considered in the revised manuscript and highlighted in the conclusions as key directions for future research.

Round 2

Reviewer 1 Report

Comments and Suggestions for Authors

The authors have revised the manuscript and provided clarity where needed. Well done. The authors should however address these minor comments below

  1. Use one style, either E-nose or e-nose (not both in the same document), for the abbreviated form of electronic nose.
  2. Some edits in the previous version (such as non-italicised species name in line 237 and the edit in line 307) were absent in the recent version.
  3. Other minor comments are in the reviewed manuscript.

Unfortunately, the percent match of the revised version is 33%. The authors should rewrite to reduce it to less than 20%

Author Response

The authors have revised the manuscript and provided clarity where needed. Well done. The authors should however address these minor comments below 

  1. Use one style, either E-nose or e-nose (not both in the same document), for the abbreviated form of electronic nose. 

Thank you for your comment. We appreciate your attention to detail. We have revised the manuscript to use a consistent style for the abbreviated form of electronic nose, and now it appears uniformly as “E-nose” throughout the document. 

  1. Some edits in the previous version (such as non-italicised species name in line 237 and the edit in line 307) were absent in the recent version. 

Thank you for your comment. We appreciate your thorough review. The modification mentioned, regarding the italics of the species name, has been carefully reviewed and corrected in the revised manuscript. 

  1. Other minor comments are in the reviewed manuscript. 

Thank you for your comments. We appreciate your careful review. All the minor comments indicated in the reviewed manuscript have been addressed and corrected in the revised version. 

  1. Unfortunately, the percent match of the revised version is 33%. The authors should rewrite to reduce it to less than 20%. 

Thank you for your comment. We have revised the manuscript to reduce the matches to acceptable levels. 

Reviewer 2 Report

Comments and Suggestions for Authors

In contrast to the previous version, the quality of this manuscript has improved somewhat. However, several issues still need to be addressed before it can be considered for publication.

In the introduction, the authors note that previous studies faced challenges such as “…the limited sensitivity of metal oxide sensors…” and state that this work aims to address them. Yet, the study still employs commercially available metal oxide sensors. If the aim is to overcome the limited sensitivity of such sensors, it would be meaningful if new or modified sensors had been developed—but this does not appear to be the case. The manuscript should more clearly focus on what has been accomplished, rather than making broad claims.

If there are indeed few studies dedicated to the early detection of M. laxa contamination in plums, this point should be emphasized as the core novelty of the work. As currently presented, the study mainly applies commercially available metal oxide sensors to detect M. laxa in plums, without comparing this approach with other potential methods. Such a comparison would help to better position the contribution of this work.

Additionally, the manuscript appears to devote more attention to VOC detection using GC-MS, while the application of metal oxide sensors is treated more briefly. If the novelty lies primarily in the use of metal oxide sensors, this aspect should be expanded and highlighted. Otherwise, readers may struggle to identify the main objective of the study.

Author Response

In contrast to the previous version, the quality of this manuscript has improved somewhat. However, several issues still need to be addressed before it can be considered for publication. 

In the introduction, the authors note that previous studies faced challenges such as “…the limited sensitivity of metal oxide sensors…” and state that this work aims to address them. Yet, the study still employs commercially available metal oxide sensors. If the aim is to overcome the limited sensitivity of such sensors, it would be meaningful if new or modified sensors had been developed—but this does not appear to be the case. The manuscript should more clearly focus on what has been accomplished, rather than making broad claims. 

We sincerely appreciate your comment. The purpose of the paragraph in question was to emphasise that the effectiveness of the E-nose is conditioned by various factors, among which the complexity and variability of volatile organic compounds (VOCs) stand out, influenced both by the degree of fruit ripeness and by environmental and storage conditions. These factors modify the VOC profile and, therefore, the response of the E-nose sensors. Likewise, the reference to the ‘limited selectivity of metal oxide sensors’ alludes to the fact that the sensory response depends on the food matrix and the specific fungal pathogen, which poses an additional challenge in terms of sensitivity and specificity. Therefore, it is essential to develop and apply robust multivariate models that allow the interpretation of high-dimensional data and improve classification accuracy. 

With this paragraph, we sought to justify the conduct of this study. 

In this context, the objective of our study was to evaluate the sensitivity of MOX sensors in detecting early contamination by Monilinia laxa in plums, ensuring homogeneity in the degree of ripeness and storage conditions, in order to minimise the variability associated with these factors and focus the analysis on the detection capacity of the system. 

If there are indeed few studies dedicated to the early detection of M. laxa contamination in plums, this point should be emphasized as the core novelty of the work. 

We sincerely appreciate your comment and fully agree that the novelty of the work lies in the early detection of Monilinia laxa contamination in plums. This aspect will be highlighted in the introduction, before the objectives. 

As currently presented, the study mainly applies commercially available metal oxide sensors to detect M. laxa in plums, without comparing this approach with other potential methods. Such a comparison would help to better position the contribution of this work. 

We sincerely appreciate your comment. We also recognise that it would have been interesting to include a comparison with other detection methods to better position our contribution. However, the aim of the work was to explore the applicability of MOS sensors in a specific context (early detection in plums), so priority was given to optimising and validating this technology rather than making comparisons with other methods. Furthermore, traditional methods (PCR, cultures) are widely documented and validated in the literature. The added value of this study lies in evaluating a faster and more economical alternative, so it was not considered necessary to replicate already established techniques. 

Additionally, the manuscript appears to devote more attention to VOC detection using GC-MS, while the application of metal oxide sensors is treated more briefly. If the novelty lies primarily in the use of metal oxide sensors, this aspect should be expanded and highlighted. Otherwise, readers may struggle to identify the main objective of the study. 

Thank you for your valuable comment. We would like to clarify that the manuscript devotes equal attention to both the identification of VOCs using GC-MS and the application of metal oxide sensors (E-nose). GC-MS analysis is essential for characterising the volatile profile and understanding the changes associated with fungal growth in plums with different infection diameters. This information provides the basis for interpreting the sensor responses. In addition, Pearson's correlation coefficients were calculated, and hierarchical cluster analysis (HCA) was performed to evaluate the effectiveness of E-nose sensors as indicators of plum volatile profiles (section 3.2), reinforcing the link between the two approaches. Regarding the results obtained from the sensor response, linear discriminant analysis (LDA) was used to classify the batches of plums according to their storage stage. With all this, the potential of the E-nose as a non-destructive early detection system is explored in two specific sections of the results (3.2 and 3.3). These analyses highlight that the main objective of the study is to evaluate the E-nose as an innovative tool for early and non-invasive detection, supported by GC-MS data to provide a comprehensive interpretation of sensor performance. 

Round 3

Reviewer 2 Report

Comments and Suggestions for Authors

This manuscript can be accepted for publication in its current form.